# Plant-Derived Sulforaphane Suppresses Growth and Proliferation of Drug-Sensitive and Drug-Resistant Bladder Cancer Cell Lines In Vitro

**DOI:** 10.3390/cancers14194682

**Published:** 2022-09-26

**Authors:** Hui Xie, Jochen Rutz, Sebastian Maxeiner, Timothy Grein, Anita Thomas, Eva Juengel, Felix K.-H. Chun, Jindrich Cinatl, Axel Haferkamp, Igor Tsaur, Roman A. Blaheta

**Affiliations:** 1Department of Urology and Pediatric Urology, University Medical Center Mainz, 55131 Mainz, Germany; 2Department of Urology, Goethe-University, 60590 Frankfurt am Main, Germany; 3Institute of Medical Virology, University Hospital, Goethe-University, 60596 Frankfurt am Main, Germany

**Keywords:** sulforaphane, bladder cancer, drug resistance, proliferation, AKT/mTOR

## Abstract

**Simple Summary:**

The natural compound sulforaphane is highly popular among tumor patients, since it is suggested to prevent oncogenesis and cancer progression. However, knowledge about its precise mode of action, particularly when drug resistance has been established, remains poor. The present study demonstrates the proliferation-blocking effects of SFN on a panel of drug-resistant bladder cancer cell lines.

**Abstract:**

Combined cisplatin–gemcitabine (GC) application is standard for treating muscle-invasive bladder cancer. However, since rapid resistance to treatment often develops, many patients turn to supplements in the form of plant-based compounds. Sulforaphane (SFN), derived from cruciferous vegetables, is one such compound, and the present study was designed to investigate its influence on growth and proliferation in a panel of drug-sensitive bladder cancer cell lines, as well as their gemcitabine- and cisplatin-resistant counterparts. Chemo-sensitive and -resistant RT4, RT112, T24, and TCCSUP cell lines were exposed to SFN in different concentrations, and tumor growth, proliferation, and clone formation were evaluated, in addition to apoptosis and cell cycle progression. Means of action were investigated by assaying cell-cycle-regulating proteins and the mechanistic target of rapamycin (mTOR)/AKT signaling cascade. SFN significantly inhibited growth, proliferation, and clone formation in all four tumor cell lines. Cells were arrested in the G2/M and/or S phase, and alteration of the CDK–cyclin axis was closely associated with cell growth inhibition. The AKT/mTOR signaling pathway was deactivated in three of the cell lines. Acetylation of histone H3 was up-regulated. SFN, therefore, does exert tumor-suppressive properties in cisplatin- and gemcitabine-resistant bladder cancer cells and could be beneficial in optimizing bladder cancer therapy.

## 1. Introduction

Accounting for approximately 570,000 cases and 210,000 deaths per year, bladder cancer reflects the tenth most common cancer worldwide. A significant gender difference is present in this cancer, being nearly four times more prevalent in men than in women [1]. High recurrence rates and progression to higher tumor stages make it a life-threatening disease, requiring lifelong surveillance.

Current therapeutic management depends on the histology, whereby three subtypes have been broadly defined: non-muscle-invasive bladder cancer (NMIBC), muscle-invasive bladder cancer (MIBC), or metastatic bladder cancer [2]. Once spread into the muscularis propria, adjacent organs, or the pelvic or abdominal wall, the disease is difficult to treat, due to the rapid development of systemic micrometastases. A cisplatin–gemcitabine (GC) combination therapy is the most widely applied approach for these patients [3]. However, the latest randomized phase III trial [4] showed that only 36% of MIBC patients receiving GC therapy responded adequately. Despite the initially high response rate, resistance rapidly develops as intracellular growth signaling pathways become reactivated. The median survival rate of patients undergoing a cisplatin-based regimen is only 14 months [5].

Effective second-line treatment options have not been developed to overcome failure or resistance to GC therapy. Immune checkpoint inhibitors targeting programmed death 1 (PD1) and programmed death-ligand 1 (PD-L1) have recently revolutionized the therapeutic paradigm of platinum-refractory bladder carcinoma [6]. Nevertheless, only a small subset of patients respond to this form of therapy [7]. Due to acquired resistance or severe side effects, patients dissatisfied with conventional cancer treatment are increasingly turning to complementary alternative medicine (CAM) [8]. Around 50% of cancer patients worldwide presently use CAM during the course of their disease, whereby administration of natural herbs is the preferred CAM modality [9,10].

Integrating plant-derived compounds into conventional anticancer treatment protocols is hoped to result in cancer regression and prevent cancer recurrence. Though a majority of patients are satisfied with applied CAM methods, many CAM compounds have not been well studied, and evidence-based trials are lacking [11,12]. The present study deals with the relevance of the natural isothiocyanate sulforaphane (SFN) as a beneficial CAM. The precursor of SFN, glucoraphanin, is enriched in cruciferous vegetables such as broccoli, cauliflower, and cabbage. Indeed, the consumption of raw cruciferous vegetables has been shown to reduce the risk of bladder cancer occurrence and to improve the survival of bladder cancer patients [13,14]. To exert tumor-protective properties, glucoraphanin, which itself is not bioactive, requires hydrolytic conversion into SFN by myrosinase. This enzyme is present in the plant tissue but also enriched in the gastrointestinal microflora. Since SFN was isolated from broccoli in 1992, evidence has been provided that this compound exerts multiple effects in cancer cells, with histone deacetylase (HDAC) inhibition being one potential mechanism [15,16]. This is relevant since alteration of HDACs has been closely associated with a wide range of diseases including various cancers. Indeed, alteration of the histone acetyltransferase (HAT)/HDAC balance towards elevated histone deacetylation and enhanced transcription of tumor suppressor genes is well documented in bladder cancer [17]. Consequently, pharmacological targeting of HDACs to counteract aberrant cell growth activity has emerged as a potential strategy to treat bladder cancer.

Whether SFN acts as a natural HDAC inhibitor in bladder cancer, particularly when cisplatin or gemcitabine resistance has occurred, is not clear. In a murine bladder cancer xenograft model, gavaging over two weeks with SFN (52 mg/kg body weight/day) was shown to reduce tumor weight by 42%, compared to the controls. This effect was concomitant with down-regulated HDAC activity [18,19]. SFN has also been demonstrated to reverse chemotherapeutic resistance in kidney cancer cell lines, and no resistance has been observed during the chronic application of SFN [20,21]. Based on the reportedly beneficial characteristics of SFN, the present study was conducted to evaluate SFN’s potential in diminishing growth and proliferation in a panel of drug-sensitive, cisplatin- and gemcitabine-resistant bladder cancer cell lines.

## 2. Materials and Methods

### 2.1. Cell Culture and Resistance Induction

RT4, RT112, T24 (ATCC/LGC Promochem GmbH, Wesel, Germany), and TCCSUP (DSMZ, Braunschweig, Germany) bladder carcinoma cells (sensitive) and their corresponding cisplatin- and gemcitabine-resistant cells were investigated. Sensitive cell lines were termed RT4^sen^, RT112^sen^, T24^sen^, and TCCSUP^sen^. The cisplatin- and gemcitabine-resistant cells were termed RT4^cis^, RT112^cis^, T24^cis^, and TCCSUP^cis^, and RT4^gem^, RT112^gem^, T24^gem^, and TCCSUP^gem^, respectively.

Bladder cancer cells were cultivated in Isocove’s Modified Dulbecco’s Medium (IMDM; Gibco/Invitrogen, Karlsruhe, Germany) enriched with 10% fetal calf serum (FCS), 2% glutamax, and 1% penicillin/streptomycin (all: Gibco/Invitrogen) in a humidified 5% CO_2_ incubator.

Resistant RT4 cells were established by exposing the sensitive cells to stepwise increasing concentrations of cisplatin and gemcitabine (both provided by Hexal, Holzkirchen, Germany), up to 2 µg /mL and 20 ng/mL, respectively. Parental RT112 and T24 cells were treated with cisplatin or gemcitabine, up to 1 µg/mL or 20 ng/mL, respectively. Resistant TCCSUP cells were established by exposing the sensitive cells over 6 months to stepwise increasing concentrations of cisplatin or gemcitabine, up to maximum concentrations of 1 µg/mL or 10 ng/mL, respectively.

Resistance was checked as follows: cisplatin- or gemcitabine-containing medium was removed and replaced by a cell culture medium free of cisplatin or gemcitabine. RT4, RT112, T24, and TCCSUP cells were then incubated for 72 h without cisplatin or gemcitabine. A medium change was also carried out with the drug-sensitive cell cultures. Thereafter, all cell cultures were washed with phosphate-buffered saline (PBS; Gibco/Invitrogen) and then subjected to medium containing different drug concentrations (0.125–4 µg/mL cisplatin or 1.25–40 ng/mL gemcitabine). Drug response was then evaluated using the MTT assay as described below. Cell lines were defined to be resistant when they no longer responded to cisplatin or gemcitabine, or when the response to the drugs was strongly restricted, compared to the response of the sensitive cells. All further experiments were then conducted by comparing drug-sensitive to drug-resistant cells that were permanently exposed to 1 µg/mL (TCCSUP, RT112, T24) or 2 µg/mL cisplatin (RT4) or 10 ng/mL (TCCSUP) or 20 ng/mL gemcitabine (RT4, RT112, T24) [22].

### 2.2. Sulforaphane (SFN)

SFN (L-Sulforaphane, Biomol, Hamburg, Germany) was applied to the cell cultures (drug-sensitive and drug-resistant) at concentrations ranging from 1 to 40 μM. Controls remained untreated. Toxic effects of SFN were checked with the trypan blue dye exclusion test (Gibco/Invitrogen, Darmstadt, Germany).

### 2.3. Cell Growth Analysis

Tumor cell growth was measured using the 3-(4,5-dimethylthiazol-2-yl)-2,5-diphenyltetrazolium bromide (MTT) dye reduction assay (Roche Diagnostics, Penzberg, Germany). Both the cisplatin/gemcitabine-resistant and drug-sensitive bladder cancer cells were placed into 96-well tissue culture plates at a concentration of 1 × 10^5^ cells/mL (50 μL/well). SFN was also added at different concentrations (controls were without SFN). After a 24, 48, and 72 h incubation, each well was filled with 10 μL MTT (0.5 mg/mL) for an incubation period of 4 h. Lysis buffer containing 10% SDS in 0.01 M HCl was then added to the cells. Following an overnight incubation at 37 °C, 5% CO_2_, absorbance at 550 nm was measured in each well using a microplate enzyme-linked immunosorbent assay reader (ELISA; Tecan Infinite M200, Männedorf, Switzerland). Each experiment was performed in triplicate. After subtracting the background absorbance and offsetting with a standard curve, the results are expressed as the mean cell number. To illustrate the dose–response kinetics, the mean cell number after 24 h incubation was set to 100%.

### 2.4. Apoptosis Detection

Fluorescent detection of annexin V served to investigate apoptotic events caused by SFN treatment. The expression was explored in drug-sensitive as well as in drug-resistant cells, whereby the annexin V-FITC Apoptosis Detection kit was used (BD Pharmingen, Heidelberg, Germany). All bladder cancer cell lines were washed twice with PBS and then incubated with 5 μL of annexin V-FITC and 5 μL of propidium iodide (PI) in the dark for 15 min at room temperature. Thereafter, stained cells were subjected to flow cytometry (FACScalibur; BD Biosciences, Heidelberg, Germany). CellQuest software (BD Biosciences) served to calculate the percentage of early and late apoptotic cells, as well as that of necrotic and vital cells.

### 2.5. BrdU Incorporation

The proliferative activity of the tumor cells in the presence of SFN was explored using the BrdU (Bromodeoxyuridine) cell proliferation ELISA kit (Calbiochem/Merck Biosciences, Darmstadt, Germany). The drug-sensitive bladder cancer cells and their respective drug-resistant counterparts were exposed to SFN at different concentrations for an incubation period of 24 and 48 h. Controls did not receive SFN. A total of 5000 cells were then transferred to each well in 96-well plates (in triplicate) and incubated thereafter with BrdU for 24 h. Tumor cells were finally fixed and immunolabeled according to the instructions of the manufacturer.

### 2.6. Clonogenic Growth

The clonogenic growth assay was carried out to detect the proliferation capacity of single cells. Bladder cancer cells were transferred to 6-well plates at 200–500 cells per well, depending on the cell line used. Then, 2 mL cell culture medium was additionally added (control medium versus SFN-containing medium). Based on the different growth rates, the plates were incubated at 37 °C for 5–11 days, and the number of colonies formed (more than 50 cells) was then counted.

### 2.7. Cell Cycle Analysis

Subconfluent drug-sensitive and drug-resistant bladder cancer cells were exposed to 20 µM SFN for 24 h. Controls remained untreated. To allow cell cycle analysis, treated and non-treated tumor cells were stained with the dye PI, using a Cycle TEST PLUS DNA Reagent Kit (BD Biosciences, Heidelberg, Germany), and then subjected to flow cytometry with a FACScan flow cytometer (BD Biosciences). A total of 10,000 events per sample were analyzed. Data acquisition was conducted with Cell-Quest software. The cell cycle distribution was calculated using ModFit software (BD Biosciences). The number of gated cells in the G1, G2/M, or S phase was finally depicted as the percentage of the total number of cells in all phases.

### 2.8. Western Blot Analysis

The protein expression of proteins involved in cell cycle regulation was explored as well in RT4, RT112, T24, and TCCSUP cells (treated with 20 µM SFN for 24 h versus non-treated, both resistant and sensitive cells). The tumor cell lysates were applied to a 7–12% polyacrylamide gel (gel composition depended on the protein size to be detected) and electrophoresed for 90 min at 100 V. Subsequently, the proteins were transferred to nitrocellulose membranes (1 h, 100 V) and blocked with nonfat dry milk for 1 h. In the next step, the membranes were incubated overnight with monoclonal antibodies. These were directed against the CDK–cyclin axis and the AKT/mechanistic target of rapamycin (mTOR) pathway: anti-CDK1/Cdc2 (IgG1, clone 1), anti-pCDK1/Cdc2 (IgG1, clone 44/CDK1/Cdc2 (pY15)), anti-CDK2 (IgG2a, clone 55), anti-cyclin A (IgG1, clone 25), anti-cyclin B (IgG1, clone 18; all: BD Pharmingen), anti-mTOR (clone 7C10), anti-pmTOR (clone D9C2), anti-Raptor (clone 24C12), anti-pRaptor (clone Ser 792), anti-Rictor (clone D16H9), anti-pRictor (clone Thr1135, clone D30A3; all: New England Biolabs), anti-PKBα/AKT (IgG1 clone 55), anti-pAKT (IgG1, Ser472/Ser473, clone 104A282; both: BD Pharmingen). To evaluate epigenetic modifications, acetyl-histone H3 (aH3) was detected by an antibody directed against aH3 (IgG, Lys9; Cell signaling, Leiden, The Netherlands). HRP-conjugated goat anti-mouse IgG and HRP-conjugated goat anti-rabbit IgG (both: 1:5000; Upstate Biotechnology, Lake Placid, NY, USA) served as the secondary antibodies. For protein visualization, the membranes were incubated with ECL detection reagent (ECL; Amersham/GE Healthcare, München, Germany), and then protein bands were analyzed using the Fusion FX7 system (Peqlab, Erlangen, Germany). β-Actin (1:1000; clone AC-15; Sigma-Aldrich, Taufenkirchen, Germany) served as the internal control. To quantify the intensity of the protein bands, the protein intensity/β-actin intensity ratio was calculated with GIMP 2.8 software.

### 2.9. Statistics

All experiments were carried out three to six times. Statistical significance was calculated with the Wilcoxon–Mann–Whitney U test or a *t*-test. Differences were considered statistically significant at a *p* value less than 0.05.

## 3. Results

### 3.1. Resistance Induction

All cell lines were treated with increasing dosages of cisplatin or gemcitabine, and drug efficacy was evaluated after 3 to 6 months using the MTT assay. The tumor cell response to drug treatment is shown in Figure 1 (sensitive versus resistant cells). TCCSUP^sen^, RT112^sen^, and RT4^sen^ already responded to 1.25 ng/mL gemcitabine. Concentrations of >2.5 ng/mL gemcitabine were necessary to suppress the growth of T24^sen^ cells. In contrast, 1.25 and 2.5 ng/mL gemcitabine did not induce growth suppression (TCCSUP^gem^, RT112^gem^), and these concentrations elevated the tumor cell number (T24^gem^, RT4^gem^), compared to the untreated controls. A distinct response of the resistant sublines to drug treatment was only observed at 40 ng/mL gemcitabine. Cisplatin >0.125 µg/mL (TCCSUP^sen^, RT112^sen^, RT4^sen^) or >0.5 µg/mL (T24) significantly reduced the tumor growth of the sensitive cell lines, whereas higher concentrations of cisplatin were required to (moderately) diminish the cell number in the resistant cell lines (RT4^cis^: >0.5 µg/mL; TCCSUP^cis^, T24^cis^, RT112^cis^: >2 µg/mL).

### 3.2. SFN Blocks Growth of Drug-Sensitive and Drug-Resistant Bladder Cancer Cells

SFN blocked the growth of both the drug-sensitive and drug-resistant bladder cancer cell lines (Figure 2). However, efficacy depended on the cell line and resistance status. RT112^sen^ already responded to 1 µM SFN, whereas concentrations of >5 µM (RT112^cis^) and >20 µM (RT112^gem^) were necessary to significantly reduce the cell number of the respective resistant sublines. The cell number of TCCSUP^sen^, TCCSUP^gem^, and TCCSUP^cis^ as well as that of T24^gem^ and T24^cis^ was down-regulated in the presence of >10 µM SFN. However, T24^sen^ also responded to 5 µM SFN. The same concentration range (>5 µM SFN) diminished the cell number of RT4^sen^, RT4^gem^, and RT4^cis^, all related to the controls. The trypan dye exclusion test did not reveal signs of toxicity.

### 3.3. Apoptosis Induction by SFN

Apoptosis was evaluated in all cell lines. Representative data are shown for TCCSUP and RT112 cells (sensitive and resistant) in Figure 3, following a 24 h drug incubation with SFN (0 (controls, ctrl), 15, 20, and 30 µM). In the presence of 15 µM SFN, a moderate in-crease in late apoptosis in RT112 cells (sensitive, cisplatin-resistant, gemcitabine-resistant) was observed. Both late and early apoptosis slightly increased in TCCSUPcis as well. Maximum effects on early apoptosis were seen when the sensitive, cisplatin-resistant, or gem-citabine-resistant tumor cells were exposed to 30 µM SFN (RT112 > TCCSUP). A moderate increase in necrotic cells was also seen when RT112gem cells were treated with 30 µM SFN.

### 3.4. BrdU Incorporation

To evaluate the influence of SFN on tumor cell proliferation, the BrdU incorporation assay was employed. Figure 4 shows the data after a 48 h incubation period. Potent effects of SFN were seen when it was applied at 20 and 30 µM. Effects were similar between sensitive and resistant cells, except for RT4, where BrdU incorporation was reduced in RT4^gem^ to a lesser extent than in RT4^sen^ and RT4^cis^.

### 3.5. Suppression of Clonogenic Tumor Growth

To investigate the potential of SFN to stop colony formation, the clonogenic growth assay was performed. Figure 5 depicts the number of clones counted and provides pictures of the cellular morphology (representative of sensitive tumor cells treated with 15 µM SFN). SFN at concentrations of >5 µM significantly reduced the number of tumor clones, except for TCCSUP^cis^, where 5 µM SFN was without effect (compared to the respective control). Morphologic analysis demonstrated compact and dense clone structures in the untreated cell lines. Disintegration of the tumor cell clones became obvious in the presence of 15 µM SFN (particularly in T24 and RT4 cells).

### 3.6. Influence of SFN on Cell Cycling

Distinct cell cycle alterations were evoked by SFN. A dose-dependent increase was apparent in the G2/M phase in TCCSUP, T24, and RT112 cells (all sensitive and gemcitabine- and cisplatin-resistant cells). Up-regulation of G2/M-phase cells was accompanied by a loss of G0/G1-phase cells, except for TCCSUP^gem^, where the number of S-phase cells was diminished, compared to the untreated control. In RT4 cells, SFN was moderately effective. When applied at 30 µM, it caused up-regulation of S-phase RT4^sen^, RT4^gem^, and RT4^cis^ cells, along with a diminished number of G0/G1-phase cells (Figure 6).

### 3.7. Evaluation of Cell-Cycle-Regulating Proteins

No homogenous tendency in parental and cisplatin- and gemcitabine-resistant bladder cancer cells was apparent in regard to protein alteration in the presence of 20 µM SFN (versus untreated controls, Figure 7). In TCCSUP cells, pCDK1 increased in the sensitive and resistant sublines, whereas CDK2 decreased in TCCSUP^sen^ under SFN. Both cyclins A and B were elevated in the sensitive and resistant cells by SFN. Interestingly, AKT decreased but pAKT increased in TCCSUP^sen^, TCCSUP^cis^, and TCCSUP^gem^. The proteins mTOR, pmTOR, and pRaptor were all down-regulated by SFN in TCCSUP (pmTOR not in TCCSUP^sen^). However, pRictor increased in TCCSUP^sen^ and TCCSUP^cis^ cells. Acetylated H3 was elevated in TCCSUP^sen^ and TCCSUP^cis^ cells. Like TCCSUP, pCDK1 was enhanced in sensitive and resistant T24 cells, and cyclins A and B were up-regulated in T24^cis^ and T24^gem^. Not only AKT but also pAKT was lowered in T24^sen^ and T24^cis^ cells. A decreased expression of mTOR and Raptor (both T24^cis^, T24^gem^) as well as Rictor (T24^sen^, T24^cis^, T24^gem^) was also induced by 20 µM SFN. Phosphorylated proteins could not be detected by the respective antibodies. Acetylated H3 was elevated in all T24 cell lines. Similar to TCCSUP and T24, pCDK1 became enhanced in RT112^sen^ and RT112^cis^ by SFN (no signal in RT112^gem^). Cyclins A and B were elevated in RT112^sen^ and RT112^cis^, whereas CDK2 was suppressed in all cell sublines. AKT was diminished in RT112^cis^, and pAKT was diminished in RT112^sen^ and RT112^cis^ (no signal in RT112^gem^). Loss of mTOR (RT112^sen^, RT112^cis^) and pmTOR (RT112^gem^, not detectable in RT112^sen^ and RT112^cis^) was also evoked by SFN. In addition, Raptor (all sublines), pRaptor (RT112^cis^, not detectable in RT112^sen^ and RT112^gem^), Rictor (RT112^sen^, RT112^gem^), and pRictor (RT112^gem^) were diminished by SFN, whereas aH3 was elevated in all RT112 sublines. A distinct response to SFN was verified in RT4^cis^ cells with CDK1 and pCDK1 (up-regulation), and in RT4^sen^ cells with CDK1/pCDK1 (down-regulation). Down-regulation of pAKT, pmTOR, and Rictor was observed in RT4^sen^ and RT4^cis^ cells. Raptor was suppressed in all sensitive and resistant RT4 cells. Phosphorylated pRaptor could not be detected. aH3 increased in RT4^sen^ and RT4^cis^ cells, both compared to the untreated controls. Figure 8 depicts the relevant pixel density data.

## 4. Discussion

SFN strongly suppressed growth and proliferation in a panel of bladder cancer cell lines, both sensitive and resistant towards gemcitabine and cisplatin. Growth blockage was already apparent at SFN concentrations of 1 µM (RT112), 5 µM (T24, RT4), and 10 µM (TCCSUP) without toxicity. In T24, SW780, and 5637 bladder cancer cells, 20 µM SFN significantly reduced tumor growth and induced apoptosis, with no signs of toxicity [23,24]. Notably, SFN acted on cisplatin- and gemcitabine-resistant tumor cells. Other investigators have also reported beneficial effects associated with SFN. It sensitizes human cholangiocarcinoma to cisplatin [25] and counteracts resistance to the mTOR inhibitor everolimus in bladder cancer cells [26]. Chronic application of SFN continually causes a significant antitumor response in bladder [26], kidney [21], and pancreatic cancer cells [27], indicating that SFN itself does not seem to induce resistance. SFN, therefore, might be a promising candidate to support a GC treatment regimen by preventing resistance induction.

It should be kept in mind that the SFN concentrations used in the present investigation are in vitro. No data from SFN concentrations in bladder cancer patients are available as yet. The total levels of SFN metabolites in the plasma and urine of patients at risk of developing prostate cancer were 0.12 μM and 4.8 μM, respectively, following daily administration of two broccoli sprout extract capsules (200 μM SFN) [28]. Approximately 20 μM dithiocarbamates, a group of SFN metabolites, has been detected in the urine of healthy volunteers consuming 50 or 70 g/day broccoli sprouts [29,30]. Novel broccoli genotypes with increased levels of glucoraphanin have been developed that should improve the bioavailability of SFN [31]. Thus, the concentration of 20 μM SFN used in vitro in the present investigation could be clinically relevant.

Another hurdle to identifying an efficacious SFN concentration is that the sensitivity of the tumor cells to SFN differed among the cell lines. A different response to SFN was noted between RT4 and the cell lines TCCSUP, RT112, and T24 (based on the BrdU and clonogenic growth assays). Differences were also seen between sensitive and resistant cells. This may be expected in light of the stark differences in the cancer subtypes. RT4 is isolated from a non-invasive superficial cancer. RT112 is invasive (grade 2), and T24 is a grade 3 bladder cancer, whereas TCCSUP is isolated from a grade 4 transitional cell carcinoma. The efficacy of SFN in suppressing proliferation and clonogenic growth may, therefore, depend on the cancer subtype.

SFN exerted its antitumor properties via cell cycle arrest, predominately through accumulation of tumor cells in the G2/M phase (except for RT4). A shift of bladder cancer cells to the G2/M phase has also been noticed by others. Tang et al. reported a G2/M arrest of UMUC3 and T24 cell lines in the presence of SFN [32,33]. A similar response was triggered by SFN in renal and prostate cancer cells [34,35], indicating that G2/M accumulation might be a ubiquitous feature of SFN. Nevertheless, some differences are apparent and require further investigation. SFN only moderately elevated the G2/M phase in RT4^sen^ and RT4^gem^ cells and had no effect on RT4^cis^ cells. Instead, the number of RT4^cis^ S-phase cells was enhanced. SFN induced a decrease in G0/G1-phase TCCSUP^sen^ and TCCSUP^cis^ but not in TCCSUP^gem^ cells. Rather, the number of S-phase TCCSUP^gem^ cells was diminished. In contrast, S-phase RT112^sen^ and RT112^gem^ (but not RT112^cis^) became elevated under SFN. The fine-tuned modulation of cell cycling caused by SFN seems, therefore, to depend on the different characteristics of the cell lines and the type of resistance. Certainly, the role of SFN as a cell cycle regulator requires further investigation with a particular focus on drug-sensitive versus drug-resistant cells.

The progression of the G2/M and S phases in the cell cycle is predominantly controlled by CDK1 and CDK2 and their regulatory subunits, cyclin B and cyclin A [36]. Consequently, it is not surprising that SFN treatment up-regulated these molecules in the majority of the bladder cancer cell lines, in both drug-sensitive and drug-resistant cells. Enhancement of CDKs and cyclins may, at least in part, explain why the tumor cells accumulate in the G2/M phase in the presence of SFN.

Aberrant activation of the Akt/mTOR pathway is closely related to pro-survival and drug resistance properties in bladder cancer. High levels of mTOR activity are found in approximately 70% of urothelial carcinomas, implicating a key role of this pathway in these cancers [37]. As a result, targeting this pathway has been proposed to combat resistance development and fight bladder cancer progression. SFN did diminish AKT/mTOR signaling in the majority of the cell lines that were investigated. However, activation of the AKT/mTOR pathway has also been observed in the presence of SFN. Since the phosphorylated AKT/mTOR proteins, including the mTOR complexes mTORC1 (Raptor) and mTORC2 (Rictor), could not be detected in all cell lines and/or sublines, the data are difficult to interpret. The down-regulation of AKT, mTOR, Rictor, and Raptor in TCCSUP, T24, RT112, and RT4 cells corresponds well with the diminished tumor growth and proliferation capacity seen with SFN. However, there was increased pAKT expression in drug-sensitive and drug-resistant TCCSUP cells, and increased pRictor in TCCSUP^sen^ and TCCSUP^cis^ cells. Such anomalies are not uncommon. Application of the HDAC inhibitor valproic acid to PC3 or DU145 prostate cancer cells diminished pmTOR and pRaptor but simultaneously elevated pAKT and pRictor [38,39]. A recent investigation on hepatocellular carcinoma cells has demonstrated that AKT blockade enhanced the phosphorylation of AKT and Rictor [40]. The relevance of this is not yet clear, but it should be kept in mind that each mTOR complex drives specific cellular functions. Raptor (mTORC1) serves as the master regulator of bladder cancer growth and proliferation, whereas Rictor (mTORC2) is the main driving force of bladder cancer cell migration and invasion [37]. Since TCCSUP cell growth and proliferation were strongly blocked by SFN, it seems unlikely that enhancement of pAKT and pRictor reflects an escape phenomenon. However, it cannot be ruled out that activation of Rictor might be associated with the increased invasive behavior of this cell line.

Treatment with SFN elevated aH3 in all cell lines, except for TCCSUP^gem^ and RT4^gem^. This feature is clinically highly relevant, since 90% of all cancers are attributed to epigenetic modification [16]. Evaluation of HDAC expression in bladder cancer cell lines and in patient tissue, as well as analysis of data from The Cancer Genome Atlas, points to the close association between the HDAC level and disease pathogenesis [41]. Intriguingly, HDAC inhibition can synergize with immune checkpoint blockade for enhanced and long-lasting antitumor activity in bladder cancer therapy [42,43]. Lin et al. and Eto et al. observed a distinct correlation between the level of aH3, tumorigenesis of bladder cancer, and patient outcome [44,45]. The potential of the natural HDAC inhibitor SFN to enhance the aH3 protein level in bladder cancer via a specific diet is unquestionably attractive.

Several reports show that SFN influences apoptosis in cancer cells. Microscopic evaluation of HTB-9 and RT112 cells exposed to 20 µM SFN for 72 h [46], or of T24 cells exposed to 20 μM SFN for 24 h [47], revealed a strong increase in apoptotic cells. Early and late apoptosis were not analyzed separately in these experiments. A novel study conducted on T24 and SW780 cells demonstrated an increase in early apoptotic cells from 5% (control) to nearly 25% when treated with 20 µM SFN for 24 h [23]. Similar results were noted in the present study when RT112 cells (sensitive and resistant) were treated with 30 µM SFN. Lower percentage values were measured in the TCCSUP cell model, indicating that the sensitivity to SFN may depend on the cell line. Overall, the growth reduction seen under SFN may partially be due to elevated tumor cell apoptosis. How far apoptosis-related proteins are altered by SFN remains open.

## 5. Conclusions

The present investigation demonstrated significant growth- and proliferation-blocking properties of the natural isothiocyanate SFN, exerted on sensitive and cisplatin- and gemcitabine-resistant bladder cancer cells in vitro. Since SFN modulated AKT/mTOR signaling and acted on CDKs and cyclins, suppression of the AKT/mTOR pathway and alterations of the CDK–cyclin axis may contribute to SFN’s effects on tumor growth and proliferation. However, this remains speculative and requires further investigation. SFN’s mode of action is not homogeneous, with notable differences among the cell lines and between their resistant sublines. SFN is considered a promising integrative compound, particularly together with GC treatment, or when cisplatin and gemcitabine resistance has occurred. Further investigation is warranted in regard to the role of SFN in bladder cancer cell metastasis. Establishing optimal concentrations and exploring the bioavailability of SFN are also areas requiring further inquiry.

## Figures and Tables

**Figure 1 cancers-14-04682-f001:**
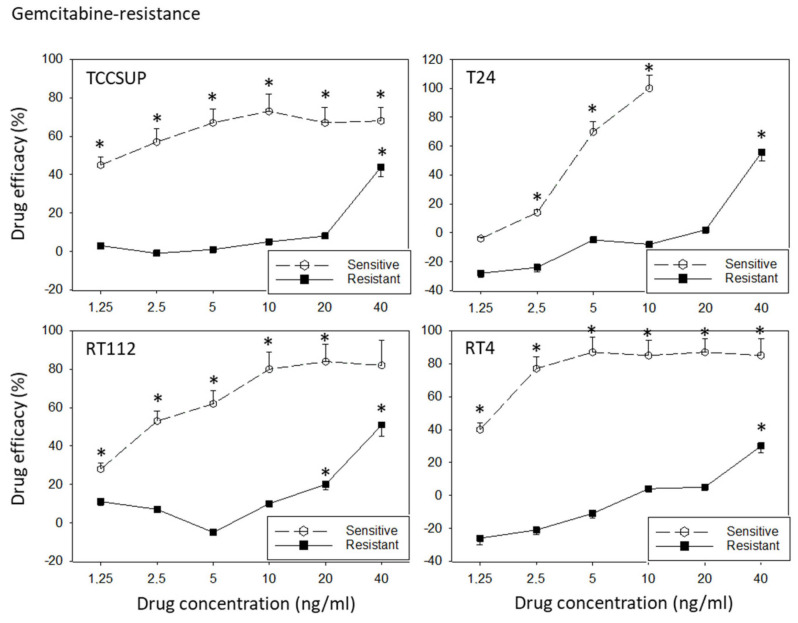
Response of sensitive versus resistant TCCSUP, T24, RT112, and RT4 bladder cancer cell lines to gemcitabine (upper graphs) and cisplatin (lower graphs). Cell lines were exposed to the compounds, the cell number was evaluated after 24, 48, and 72 h using the MTT assay, and the 24–72 h increase was determined. Differences between the cell number increase of drug-treated and non-treated (control) cells are presented as percentage drug efficacy, indicating a percentage cell number reduction. Negative values indicate an increase in the cell number under drug treatment, compared to the control. The 24–72 h cell growth data are shown in the Appendix A. Error bars indicate the standard deviation. * indicates a significant reduction in the tumor cell number, compared to the corresponding control, n = 6.

**Figure 2 cancers-14-04682-f002:**
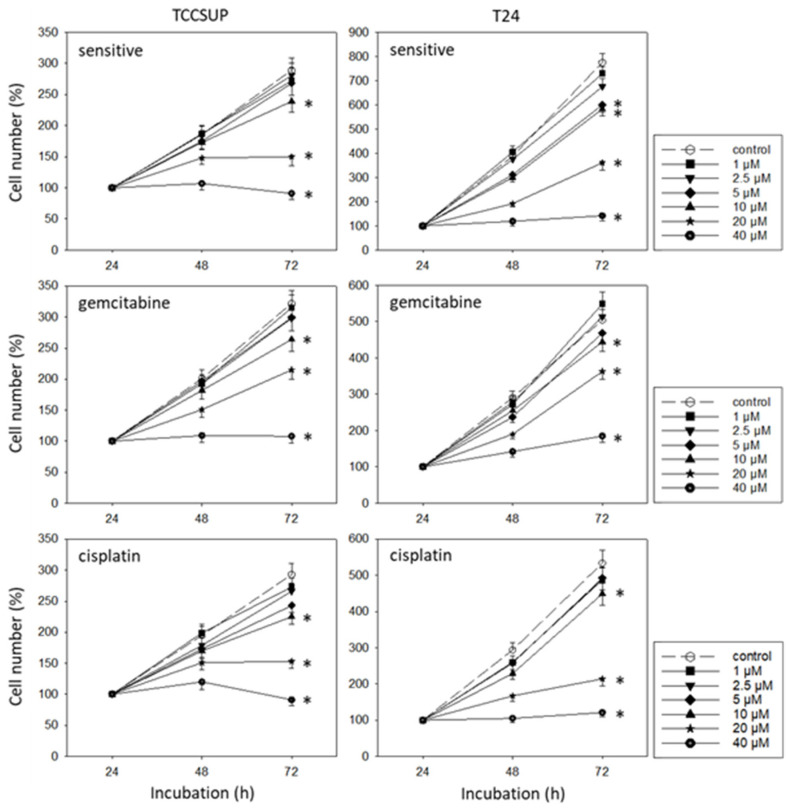
Influence of SFN (1–40 µM) on the growth of sensitive and cisplatin- and gemcitabine-resistant TCCSUP, T24, RT112, and RT4 bladder cancer cell lines. The cell number was evaluated after 24, 48, and 72 h using the MTT assay. After subtracting the background absorbance and offsetting with a standard curve, the results are expressed as the mean cell number. The mean cell number after 24 h was set to 100%. Error bars indicate the standard deviation (SD), n = 6. * indicates a significant difference compared to the non-treated control.

**Figure 3 cancers-14-04682-f003:**
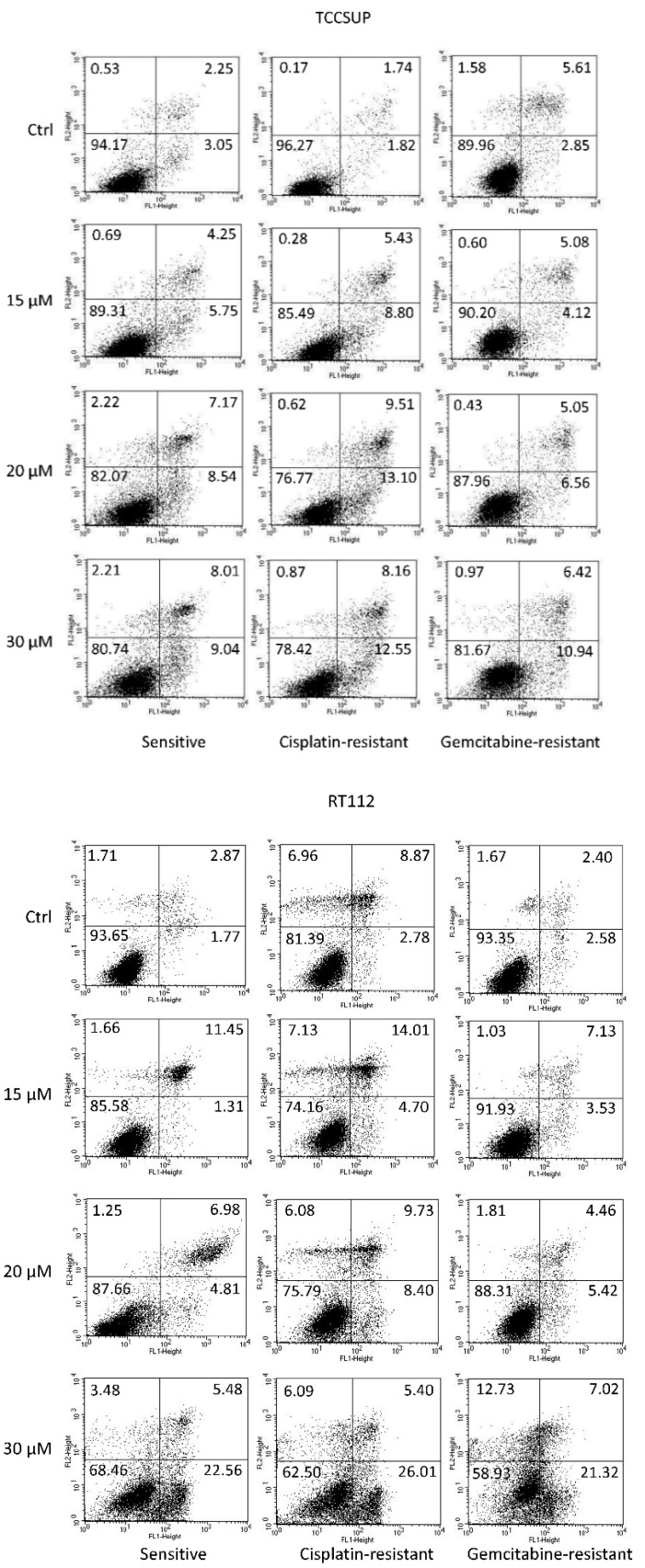
Percentage distribution of sensitive and cisplatin- and gemcitabine-resistant TCCSUP and RT112 cells undergoing early apoptosis (lower right quadrants), late apoptosis (upper right quadrants), and necrosis (upper left quadrants). Vital cells are presented in the lower left quadrants. The percentage of cells is included in each quadrant. One representative of 3 analyses (intra-assay SD <10%).

**Figure 4 cancers-14-04682-f004:**
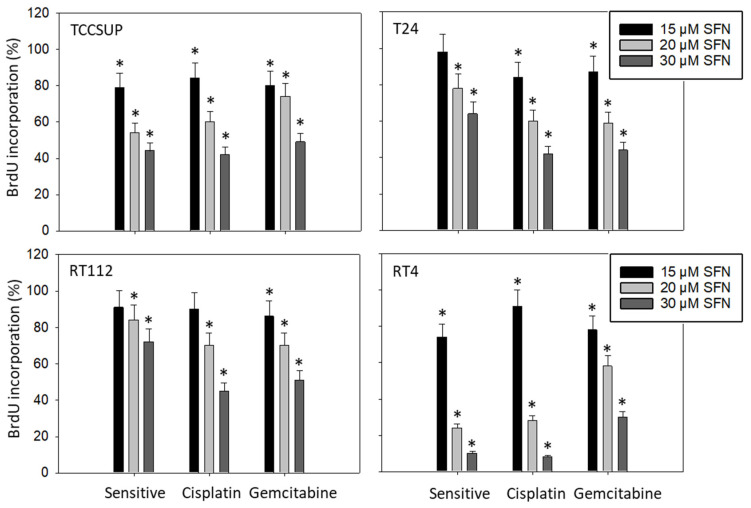
Influence of 15, 20, and 30 µM SFN on the proliferation of sensitive and cisplatin- and gem-citabine-resistant TCCSUP, T24, RT112, and RT4 bladder cancer cell lines. Evaluation by BrdU incorporation after 48 h. Error bars indicate the standard deviation. * indicates a significant difference compared to untreated controls set to 100%. n = 3.

**Figure 5 cancers-14-04682-f005:**
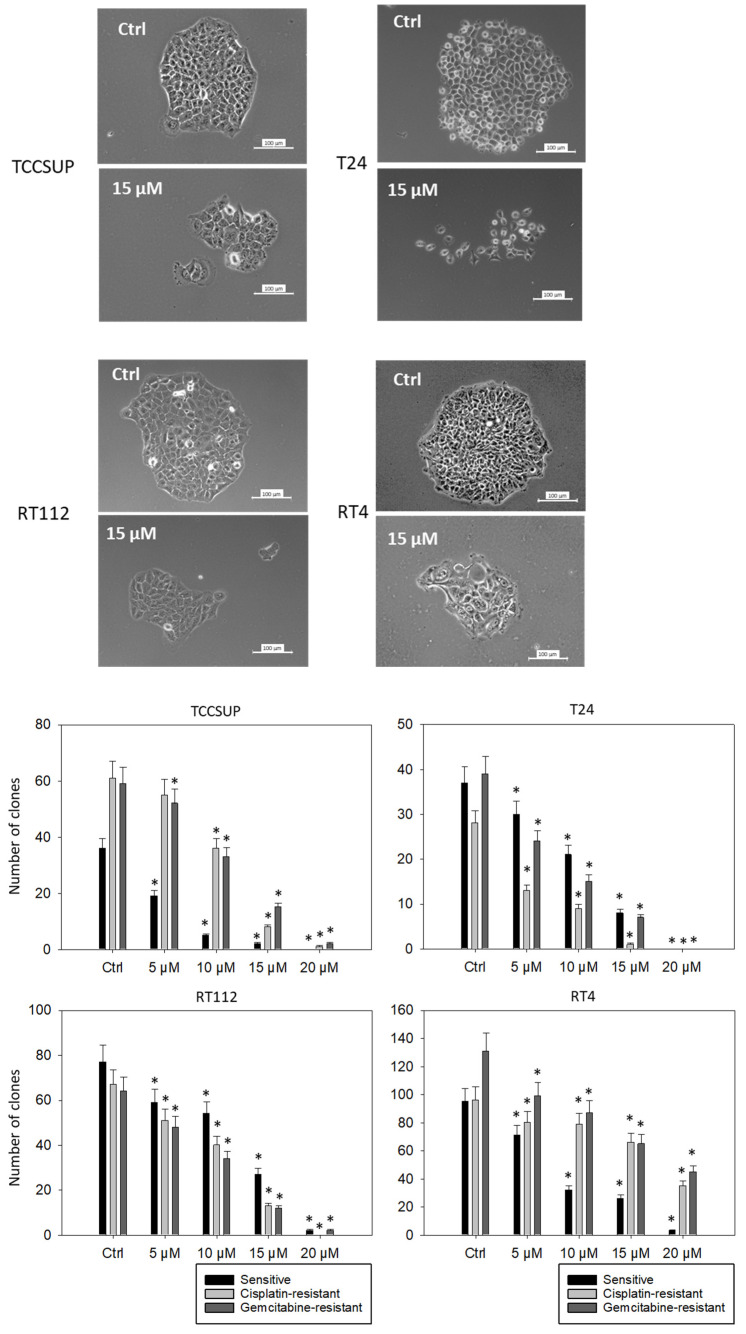
Influence of SFN on the clone formation of sensitive TCCSUP, T24, RT112, and RT4 cells. Bar diagrams indicate clone counts in the presence of 5–20 µM SFN. Pictures of single tumor clones are related to the treatment of sensitive cells with 15 µM SFN. Controls (Ctrl) were without SFN, n = 3. * indicates a significant difference compared to untreated controls.

**Figure 6 cancers-14-04682-f006:**
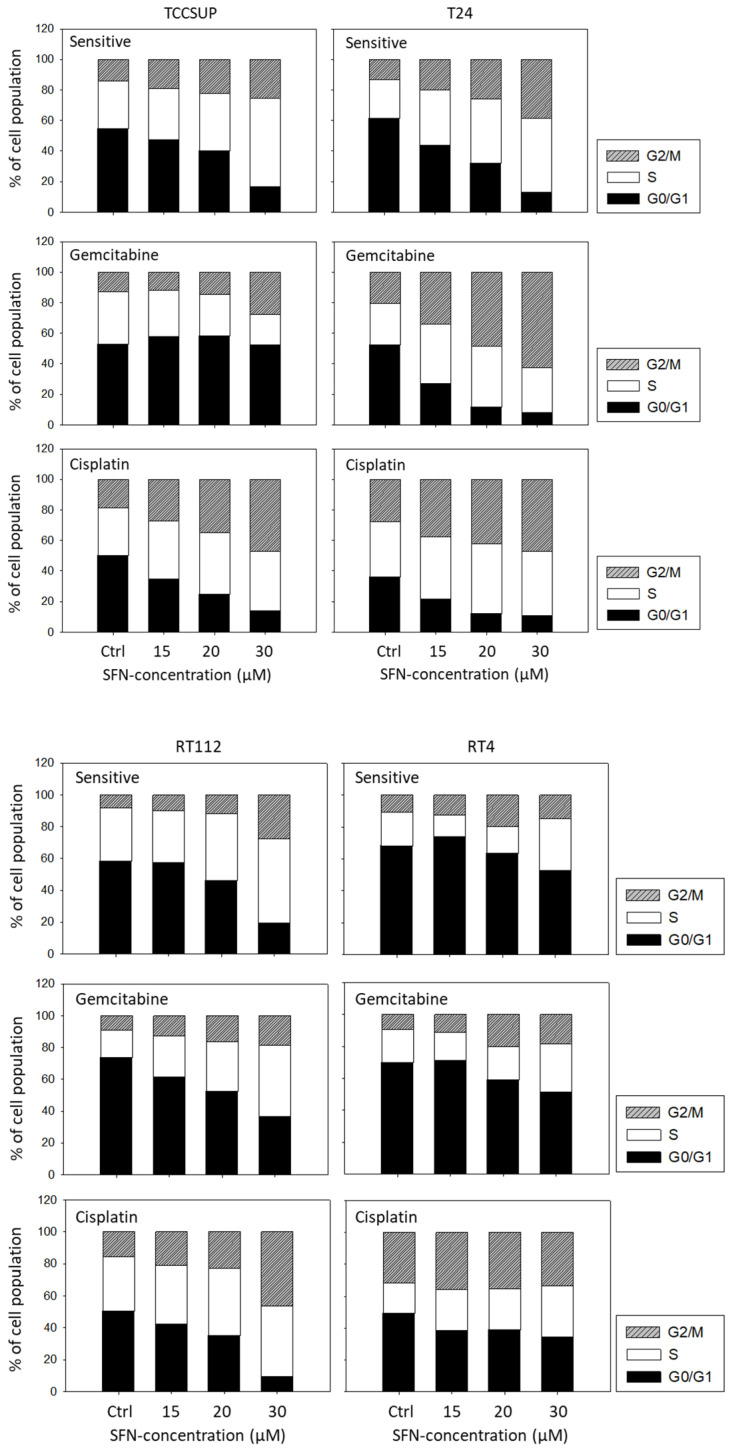
Cell cycle distribution in sensitive and cisplatin- or gemcitabine-resistant TCCSUP, T24, RT112, and RT4 bladder cancer cell lines following SFN exposure (15, 20, 30 µM). Controls (Ctrl) remained untreated. Cells in each phase are shown as a percentage. One representative of three separate experiments is shown.

**Figure 7 cancers-14-04682-f007:**
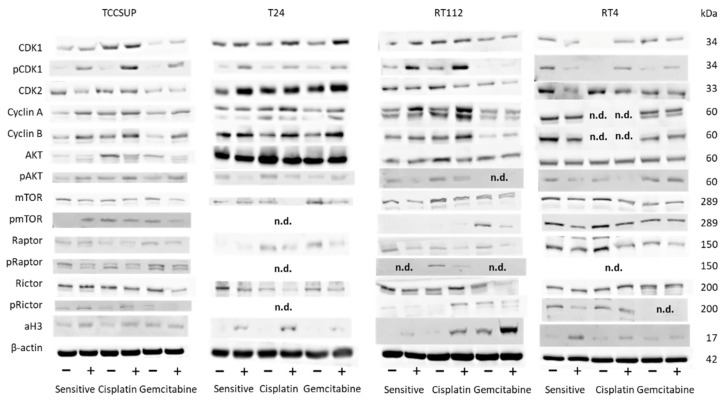
Western blot of cell cycle- and mTOR-related proteins from TCCSUP, T24, RT112, and RT4 lysates of sensitive and cisplatin- and gemcitabine-resistant cells. Tumor cells were pretreated with SFN at a concentration of 20 µM (+). Tumor cells not treated with SFN served as the controls (−). β-Actin was used as the internal control. The figure depicts representative blots from n = 3 experiments. n.d.: not detectable.

**Figure 8 cancers-14-04682-f008:**
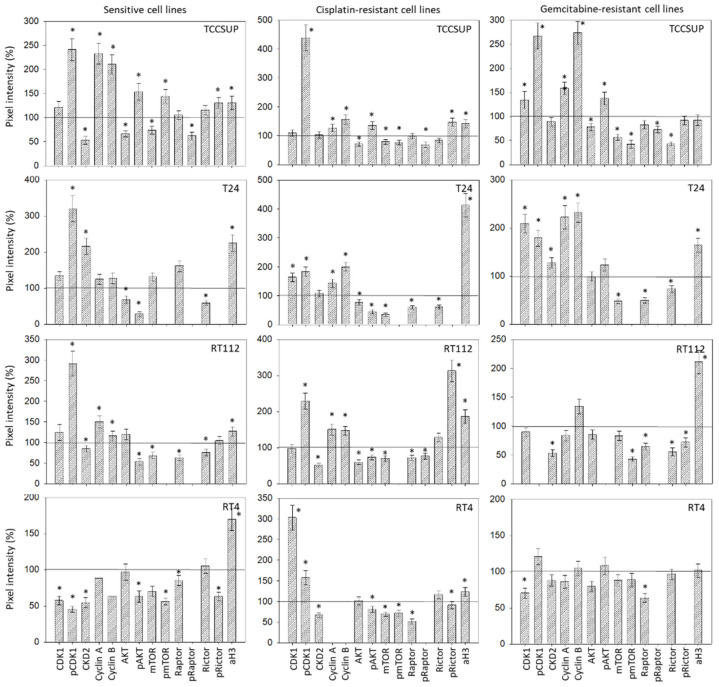
Pixel density analysis of the protein level in the sensitive and resistant bladder cancer cell lines following SFN treatment. Values are given in percentage, related to the 100% control (indicated by a black line). * indicates a significant difference compared to controls.

## Data Availability

Not applicable.

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
