# Peer review of "Plant-Derived Sulforaphane Suppresses Growth and Proliferation of Drug-Sensitive and Drug-Resistant Bladder Cancer Cell Lines In Vitro"

_cancers, 2022, doi:10.3390/cancers14194682_

Round 1

Reviewer 1 Report

Xie and co-workers presented an Original Paper for Cancers titled “Plant derived sulforaphane suppresses growth and proliferation of drug-sensitive and drug-resistant bladder cancer cell 3 lines in vitro”. Authors performed in vitro studies of growth, proliferation and cell cycle progression on bladder cancer cells. The mechanism and role of sulforaphane in inhibiting tumor progression has been investigated.  Although the results are interesting, MAJOR REVISIONS are required before to be consider this article for publication.

-       -  Statistical analysis is not understandable in most of the figures. What the authors are considering as control? This trend should be reported in figures. Are the authors assuming control viability is 100%? in this case all data should be normalized with respect to the control.

-      -   Figure 1: please substitute drug efficacy with the precise date that has been analysed (growth).

-     -    Section 3.3 must be improved.

-     -   Figure 2: MTT assay investigates cell viability. How was cell number calculated? This must be clarified.

Author Response

Comment 1+2: Statistical analysis is not understandable in most of the figures. What the authors are considering as control? This trend should be reported in figures. Are the authors assuming control viability is 100%? in this case all data should be normalized with respect to the control. Figure 1: please substitute drug efficacy with the precise date that has been analysed (growth). Our answer: We are sorry for this confusion. In figure 1, the legend has been improved and now reads, lines 221-227: “Response of sensitive versus resistant TCCSUP, T24, RT112, and RT4 bladder cancer cell lines to gemcitabine (upper graphs) and cisplatin (lower graphs). Cell lines were exposed to the compounds and the cell number was evaluated after 24, 48, and 72 h by the MTT assay, and the 24-72 h increase was determined. Differences between the cell number increase of drug treated and non treated (control) cells are presented as % drug efficacy, indicating a percent cell number reduction. Negative values indicate an increase i the cell number under drug treatment, compared to the control. The 24-72 h cell growth data are shown in supplement (S1).” We have also added the precise growth data for gemcitabine and cisplatin treated cells in a supplement (S1). In figure 2, the data are indeed related to the 24 h cell number which was set to 100%. This was done because the tumor cell attachment rate can differ when plated out into the 96-well plates. Therefore, to exclude effects related to adhesion properties and allow an exact evaluation of the tumor growth characteristics, it was necessary to normalize the cells. Unfortunately, this important detail was not mentioned in the figure legend. It has been corrected. We have now included, lines 255-257: “After subtracting background absorbance and offsetting with a standard curve, results are expressed as mean cell number. The mean cell number after 24 h was set to 100%”. We have added to the figure 6 legend, line 302: “Cells in each phase are shown as a percentage”. Comment 3: Section 3.3 must be improved. Our answer: We have now added more precise information to this section which now reads, lines 243-248: “In the presence of 15 µM SFN, a moderate increase of late apoptosis in RT112 cells (sensitive, cisplatin-resistant, gemcitabine-resistant) was observed. Both late and early apoptosis slightly increased in TCCSUPcis as well. Maximum effects on early apoptosis were seen when the sensitive, cisplatin-resistant or the gemcitabine-resistant tumor cells were exposed to 30 µM SFN (RT112 > TCCSUP).” Comment 4: Figure 2: MTT assay investigates cell viability. How was cell number calculated? This must be clarified. Our answer: We have now added in “2.3. Cell growth analysis”, lines 136-139: “After subtracting background absorbance and offsetting with a standard curve, results were expressed as mean cell number. To illustrate dose-response kinetics, the mean cell number after 24 h incubation was set to 100%.”

Reviewer 2 Report

The present study provides in vitro the anticancer effect of the natural compound sulforaphane aginst drug-sensitive and drug-resistant bladder cancer cell lines. In particular, this isothiocyanate is able to negatively affect growth and proliferation, induce apoptosis, suppress clonogenic tumor growth. Authors have exaustively demonstrated this anticancer effect by using several methods. In addition, they found that the results of the descriptive analysis was paralleled by the desensitization of AKT/mTOR pathway and alterations of the CDK-Cyclin axis. Importantly, authors used concentrations of clinical importance as motivated in the discussion. 

In my opinion, this is an interesting study, methodologically well performed. Results are clearly presented and support conclusions. The manuscript is also well discussed.

The only concern I have is related to what authors call a mechanism: "The underlying mechanism is associated with the suppression of the AKT/mTOR pathway and alterations of the CDK-Cyclin axis". Indeed, a mechanism has not benn demonstrated, since inhibitors and/or activators have not been used in the study. So, I suggest to modify the above expression and the other siilar ones that make one think of a proved true mechanism.

Author Response

Comment 1: The only concern I have is related to what authors call a mechanism: "The underlying mechanism is associated with the suppression of the AKT/mTOR pathway and alterations of the CDK-Cyclin axis". Indeed, a mechanism has not been demonstrated, since inhibitors and/or activators have not been used in the study. So, I suggest to modify the above expression and the other similar ones that make one think of a proved true mechanism. Our answer: We agree with this comment. The respective phrase in “Conclusion” has been altered and now reads, lines 445-448: “Since SFN modulated AKT/mTOR signaling and acted on CDKs and cyclins, suppression of the AKT/mTOR pathway and alterations of the CDK-Cyclin axis may contribute to SFN’s effects on tumor growth and proliferation. However, this remains speculative and requires further investigation. ” “Mechanism” has been replaced by Means in the abstract, line 24.

Round 2

Reviewer 1 Report

Authors properly revised the manuscript.